# Improving health, well-being and parenting skills in parents of children with medical complexity: a scoping review protocol

Sally Rebecca Bradshaw, Karen Shaw, Danai Bem, Carole Cummins

Institute of Applied Health Research, College of Medical and Dental Sciences, University of Birmingham, Birmingham, UK

**Correspondence to**
Sally Rebecca Bradshaw;
s.r.bradshaw@bham.ac.uk

## ABSTRACT

**Introduction** Less than 1% of children have complex medical conditions but account for one-third of all child health spending. The impact of suboptimal management of this group of children can have a considerable effect on families as well as services. Some families appear to cope more easily than others do, but there are compelling reasons to suggest that effective interventions may improve family coping and ultimately outcomes. Hospitalisation of their child presents a unique set of pressures and challenges for parents, but also an opportunity to intervene. However, the evidence is not well described in relation to this group of families. The primary objective of this scoping review is to identify parent and family-based interventions available to improve parental health, well-being, functioning or skills in the context of a child's medically complex hospital admission and hospital care.

**Methods and analysis** Nine bibliographic databases will be searched spanning medicine, nursing, psychology, education, social work and the grey literature using a combination of index terms and text words related to parents, childhood, chronic illness and interventions. Study eligibility will be assessed by two researchers against preset inclusion and exclusion criteria. Key information from each study will be extracted and charted including year of publication, condition, severity, geographical setting, key concepts and definitions, aims, study population and sample size, methodology/methods, interventions, outcomes and key findings. Directed qualitative content analysis will be used to make sense of narrative findings within the included studies. Results will be presented which summarise the scope of the literature and identify key findings, potential areas for evidence synthesis and research gaps.

**Ethics and dissemination** Ethical approval is not required. The results of this review will be disseminated through publication in a peer-reviewed journal and feedback to stakeholders during the development of a hospital-based intervention.

## BACKGROUND

This scoping review has been designed to inform development of an intervention to support parents of children with medical complexity around the time of hospital admission. It has been estimated that while children with the most complex medical needs include less than 1% of the child population, they account for one-third of all child health spending.[1] The impact of suboptimal management of this group of children, which can include the support given to enable successful family adjustment as well as medical management, can have a considerable effect on families as well as child health services and budgets.[2]

There is a well-established definition for children with special healthcare needs (CSHCN) which encompasses those children who have or are at increased risk of a chronic physical, developmental, behavioural or emotional condition and require healthcare and related services of a type or amount beyond that required by children generally.[3] Definitions for groups of children with the most severe chronic diseases or diseases with the most serious long-term effects are less well established. We have adopted the definition of 'children with medical complexity' developed by Cohen[2] which is based on a systematic review of definitions of childhood chronic conditions.[4] Cohen's definitional framework includes four domains:

► Substantial family identified service needs and/or significant impact on the family (eg, financial burden).

**Strengths and limitations of this study**

► This review will describe the literature in relation to a key opportunity to intervene to improve outcomes.
► This review will be based on a robust definition of children with medical complexity.
► Formal scoping review methodology will be used to provide a systematic, rigorous, transparent and reproducible review.
► While not a systematic review, this scoping review will be useful for identifying focused areas for systematic review.

► Diagnosed or undiagnosed chronic condition which is severe or associated with medical fragility.
► Severe functional limitations and/or dependence on technology.
► High healthcare use and/or engagement with multiple service providers that may include non-medical providers.

When describing the wider population of children with any chronic health condition, we will use the term 'CSHCN'.[3] When referencing source literature, the original terminology will be used, for example, chronic conditions, in order to retain a sense of the original meaning.

Most parents adjust to their child's illness successfully.[5 6] However, not all families do adjust well and poor adjustment has been associated with poorer health outcomes for parents, the ill child and other family members.[7] A recent meta-analysis of 37 studies where the relationship between family functioning and child well-being in children with chronic health conditions were analysed found significant correlations between family functioning and children's problem behaviours, social competence, quality of life, medication adherence and physical health.[8]

While some families appear to cope more easily than others, there are compelling reasons to suggest that effective interventions may improve outcomes for parents and their families. Some factors that predict adjustment may not be particularly open to hospital-based intervention such as family environment, illness severity and chronicity (the long-term nature of the diagnosis).[9] However, other factors that have been identified as facilitators of successful adjustment are more amenable to hospital-based intervention. These include focussing on the child's achievements, performing care routines, becoming flexible in relation to care and treatment routines, developing knowledge of the condition and treatments, being able to learn from illness episodes and apply that learning to future situations and developing effective relationships with staff.[5]

Several authors highlight the importance of the illness trajectory. Burden[10] suggests that there are opportunities for professionals to support parents to successfully adjust to their child's diagnosis. Rolland and Walsh identify three major phases of childhood chronic disease: crisis (prediagnosis and initial adjustment), chronic (the long haul) and terminal phases in progressive conditions.[11] These phases pose distinct challenges and are likely to be associated with healthcare contact and opportunities for supportive interventions to promote resilience and adjustment. The potential benefits of parenting programmes are highlighted in the findings of two reviews. A Cochrane review of group-based parenting interventions to improve parental psychosocial health found evidence to support the use of parenting programmes[12] and a separate Cochrane review found some evidence that psychological therapies are beneficial for parents of CSHCN.[13] Further evidence covering related issues have also been reviewed, for example, research on improving or supporting professional–parent collaborations in managing CSHCN,[5 14]

nursing research on parenting children with complex chronic conditions,[15] the nature of family engagement in interventions for this population[16] and the role of interactive media for parental education.[17]

However, while these reviews provide valuable insights, they do not provide a comprehensive evidence base for the context of children with medical complexity around the point of hospitalisation. Much of the available review evidence only addresses predetermined categories of interventions (eg, group,[10] psychological,[11] media[15]), and do not address other potentially important parent and family support functions such as social support, chronic illness education and skill development or support with relevant common parenting issues. In addition, they are not always well tailored to the specific parenting challenges around children with medical complexity. This review will address this knowledge gap by scoping a broad range of parent and family-based interventions that have been tested within populations of CSHCN. This is important because to our knowledge the wide range of evidence which could be relevant to the parents of medically complex children has not yet been scoped. This broad scoping review will allow that evidence to be identified, characterised and assessed in relation to the needs of these parents and families during hospital admissions and in the context of hospital care.

## METHODS/DESIGN
### Research questions and objectives
The research questions for this review are: (1) What interventions are available to improve health, well-being, functioning or skills in parents of CSHCN? (2) Who are the study populations, what were the intervention targets, which outcomes have been measured and is there evidence of efficacy or comparative effectiveness? (3) To what extent are the results relevant and transferable to delivery within routine care in a hospital setting? A further objective is to identify potential areas for full systematic review.

### Study design
Scoping review methodology is particularly well suited to this research because meeting the objectives depends on identifying and summarising a broad range of potential intervention types and research methodologies. This approach also provides a rigorous, transparent and reproducible method for scoping a research area that includes a systematic search strategy and data extraction. Formal scoping review methodology will be used,[18–22] drawing on Arskey and O'Malley's methodological framework[19] informed by recent Joanna Briggs Institute Guidance.[18] This includes identifying a research question, identifying relevant studies, study selection, charting the data and collating, summarising and reporting the results.

In order to include and describe the full extent of relevant literature, scoping reviews do not typically exclude studies based on design or quality, and data quality can

**Table 1** Population, intervention, comparator, outcome statement

| Population | Parent of children with special healthcare needs |
|---|---|
| Intervention | Any parent or family-based intervention |
| Comparator | Usual care or any other comparator |
| Outcome | Improved parenting health, well-being, functioning or skills |

therefore vary widely. The broad nature of many scoping reviews can also make study synthesis more problematic than in a full systematic review. However, both of these limitations do allow the full extent of the relevant literature to be included and described, which is useful where an area is complex or has not been comprehensively reviewed before[19] and have been addressed in this protocol.

### Eligibility criteria

The population, intervention, comparator, outcome (PICO) framework has been used to define the review focus and a PICO statement can be found in table 1. Detailed study eligibility criteria can be found in table 2.

### Search strategy

A comprehensive search strategy will be developed to identify both published and unpublished literature. It will be designed and will be performed with advice and support from a specialist in systematic reviews. A range of sources will be searched including the following disciplines: medicine, nursing, allied health professions, sociology, psychology, education and social work. Peer-reviewed, published literature will be searched as well as grey literature. Grey literature will be searched in order to increase the chance of finding evaluations that not have been published in peer-review journals. Primary research studies that evaluate interventions using any methodology and secondary research studies including scoping reviews, systematic reviews and meta-analyses will be included.

Relevant studies will be identified through individual searches of relevant data bases. These will include Medline, Embase, PsycINFO, the Cochrane library, the Cumulative Index of Nursing and Allied Health Literature (CINAHL), Education Resources Information Centre, and Applied Social Sciences Index and Abstracts. Health Management Information Consortium and Open-Grey will be searched for grey literature. Reference lists will be mined for additional references. No previous similar reviews have been found and therefore no date restrictions will be applied. Searches will be restricted to English language papers.

A phased search strategy will be used and the initial search of Medline and CINAHL will be performed using the text words shown in table 3 and related index terms.

The primary researcher will screen initial search results, abstracts of relevant studies will be retrieved and will be analysed by the same researcher for text words contained in the titles and abstracts, as well as index terms used to describe the articles. In discussion with a systematic review specialist, the results from these first stage searches will be used to optimise the search strategy for second stage searching. The second stage search will be performed individually across all databases using all identified text words and index terms found in phase 1, with search terms and strategies optimised for each database.

### Study selection

EndNote (Thomson Reuters, New York, USA) will be used to manage the records identified from the literature search and to record decisions during the study selection process. Two researchers will screen all titles from the full search results and a third researcher will take a final decision where disagreements cannot be resolved. Full texts of all potentially relevant studies will then be retrieved in full and assessed by two researchers for a final inclusion decision. Finally, reference list mining will be used to identify any further eligible studies. The selection process will be illustrated using a Preferred Reporting Items for Systematic Reviews and Meta-Analyses flow diagram.

### Data extraction, analysis and synthesis

One researcher will extract data using a prespecified data extraction form which will reflect the research questions, and this will be checked by a second researcher. Key information from each included study will be charted in a table which will include the author, year of publication, medical condition(s), severity, geographical setting, academic/professional discipline, key concepts and definitions, aims, study population and sample size, study design, methodology/methods, intervention, outcomes and key findings related to the research questions. This list is indicative only and the charting process will be iterative. As the reviewers become familiar with the evidence, the data extraction form may be updated with other headings to ensure that all relevant information is included. In addition, the risk of bias in controlled intervention studies which contain comparative information on effectiveness will be appraised using conventional systematic review methods.[23]

Directed qualitative content analysis[24] will be undertaken to analyse narrative data. Primary coding will be based on the TiDieR Framework[25] to identify author descriptions of why, what, who, how, where, when and how much, tailoring, modification and how well interventions were delivered. In terms of 'what' interventions will be coded to reflect their primary mechanism (eg, educational, psychological) and will be further coded to reflect their theoretical underpinning. Where possible more specific codes will be applied, for example, psychological interventions will be coded to reflect whether they are behavioural, cognitive or psychodynamic, etc. Data that does not fit within this approach will be identified

**Table 2** Inclusion and exclusion criteria

### Inclusion criteria

| | |
|---|---|
| Types of studies | Any reports of interventions using a recognised study design (including primary or secondary research). Interventions must aim to improve health and well-being, functioning or skills in parents of children with special healthcare needs (CSHCN). |
| Setting | Studies undertaken in any research setting (eg, acute, primary care, community) will be included, as long as the intervention could potentially be delivered within routine care in an acute setting. |
| Population | 'Parent' may include anyone with parenting responsibility. CSHCN: children who have or are at increased risk of a chronic physical, developmental, behavioural or emotional condition and require healthcare and related services of a type or amount beyond that required by children generally. |
| Intervention | Interventions must include parents directly. They may include only parents or parents alongside children and/or other family members. Interventions may include but are not limited to peer-support, listening and encouraging, education, training, enablement, modelling or environmental restructuring (eg, care environments). Single disease studies will be included (eg, cardiac conditions, cancer, metabolic conditions) as long as they meet the above inclusion criteria, as well as studies that include parents of children with a variety of clinical conditions. |
| Outcomes | Improved parent–child attachment or parenting health, well-being, functioning or skills. 'Health and well-being' may include patient-reported outcome measures, happiness, psychological adjustment or adaptation, quality of life, resilience, coping or self-efficacy. It may also include reduction in negative outcomes including stress, anxiety, depression or physical health measures. Parental functioning and skills refers to a range of parenting behaviours including nurturing, discipline, teaching, monitoring and management.[26] |

### Exclusion criteria

▲ Studies on attention deficit hyperactivity disorder, autism, depression or other mental health conditions in the absence of comorbidities.
▲ Any studies which do not report parent outcomes.
▲ Interventions that are not adaptable to delivery by generalist healthcare staff or lay workers (eg, specialist psychotherapy techniques).
▲ Studies which use parent-based interventions but only measure child well-being or disease related outcomes such as medication adherence.
▲ Studies that focus on acute conditions only (eg, acute pneumonia).
▲ Studies that focus only on end of life care.

## Table 3 Key word search terms

| Key concept | Keywords |
|---|---|
| Parents | Parent, mother, father, carer, guardian |
| Child | Child, school child, kid, toddler, teen, boy, girl, minor, underage, juvenile, youth, puberty, pubescent, prepubescent, paediatric, paediatric, school, adolescent |
| Chronic childhood disease | Chronic, long term, activity limiting, disease, illness, disorder, condition, sickness, pain |
| Interventions to improve well-being | Intervention, therapy, trial, review, meta-analysis |

and analysed to determine whether they represent a new coding category or a subcategory of an existing code.

### Presentation of results, discussion and conclusions

Results will be presented visually and descriptively. Key data will be presented in tables, including a main table of all interventions that meet the inclusion criteria. Additional data tables will summarise other key features including research methodology and design, study dates, medical condition(s), severity, geographical location, academic/professional origin and intervention function. Results of the directed qualitative content analysis will accompany the tables to further explore and discuss key findings in relation to the scoping review questions and their implications. The discussion and conclusions will also address potential areas for evidence synthesis and any identified research gaps.

### Protocol amendments

Any important amendments to this protocol will be reported with the results of this review.

### What this study will add

This study will describe the evidence base available for parenting interventions for parents of children with special healthcare need, and support development of interventions for children with medical complexity. This scoping review will contribute to a novel parent support intervention that can be delivered from within the hospital setting.

**Contributors** SRB conceived and wrote the protocol and is the guarantor of this review. KS and CC provided academic supervision and contributed to the final draft. DB provided detailed technical advice and contributed to the final draft.

**Funding** All authors are funded by the NIHR CLAHRC West Midlands initiative. This paper presents independent research and the views expressed are those of the author(s) and not necessarily those of the NHS, the NIHR or the Department of Health.

**Competing interests** None declared.

**Provenance and peer review** Not commissioned; externally peer reviewed.

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
