## [Reviewer comments · BMJ Open]

ARTICLE DETAILS

TITLE (PROVISIONAL)	Improving health, wellbeing and parenting skills in parents of children with medical complexity- A scoping review protocol
AUTHORS	Bradshaw, Sally; Shaw, Karen; Bem, Danai; Cummins, Carole

VERSION 1 – REVIEW

REVIEWER	Ruth Gilbert University College London UK
REVIEW RETURNED	26-Dec-2016

GENERAL COMMENTS	The topic of how to support parents of children with complex conditions is important and an area where more robust evidence is needed. Major comments 1. The nature of the interventions to be scoped could be made more explicit. The purpose as stated in the abstract is to improve parent-child attachment, parental health, wellbeing, functioning or skills. The abstract could be clearer about whom the interventions might be targeted at (parents or child, whole families, or health or other service providers supporting parents) and when and where the intervention might take place. On page 5 it becomes clearer that interventions for staff or services that might impact on parent or family health are excluded – the intervention needs to involve the parents.2. There is also some contradiction between the introduction and the section on the research question, where the focus seems to be on interventions that are healthcare based. The question (3) states routine health care contact. However, the inclusion criteria on page 6 still leaves it unclear as to whether non-healthcare settings (eg schools or home) will be included. For example, there is a large literature on self-help groups, which use volunteers (as in the inclusion criteria) but take place outside the health service. Will these interventions be included?3. The scoping review excludes studies where the parent has the long-term condition. In practice, a disproportionate number of parents of CSHCN themselves have disability or chronic conditions. Does this mean that interventions that improve parental functioning through addressing the parent's health needs (eg mental health care for parents) would be excluded? Is the scoping review focused on parental management of the child, rather than on parental needs or family functioning (as mentioned in the last paragraph on page 3)?4. Search terms: One of the questions stated on page 5 is "do the interventions work". To determine effectiveness, it seems important to identify studies that involve randomised comparisons. I would expect to see a more complex search strategy to achieve this.
---

	5. The section on data analysis does not include any analysis of the strength of the evidence to answer a specific question. This could be a lengthy process rather like a systematic review. However, a summary that at least captures study design is needed to indicate areas of strengths and gaps in evidence. Minor 6. Abstract para 1 last line be clearer about what is meant by “this context”
--	--

REVIEWER	Michael Takagi Murdoch Childrens Research Institute, Australia
REVIEW RETURNED	05-Apr-2017

GENERAL COMMENTS	The authors have submitted a protocol to conduct a scoping review for improving health, wellbeing, and parenting skills in parents and children with medical complexity. This is an important and interesting area and I look forward to reading the final paper. I have a few minor comments that I think will improve the protocol prior to publication. Methods/design: I think that it would be helpful to include a brief description of what formal scoping review methodology entails. Search strategy: Please explain/justify why you will include grey literature in your search. A few suggestions for additional search terms Parents: Include “guardian” Child: Include “adolescent” Interventions: Include “meta-analysis” Data analysis and synthesis; page 9; line 23: “Data will be coded according to predetermined codes derived from the research questions”. Please provide a brief example of this process. Overall the protocol is informative and well-written. Best of luck conducting the study.
--

REVIEWER	Karen Benzies Faculty of Nursing, University of Calgary, Canada
REVIEW RETURNED	28-Apr-2017

GENERAL COMMENTS	Thank you for the opportunity to review the protocol for this interesting and very important scoping review proposing to synthesize the evidence about improving health, well-being and parenting skills in parents of children with medical complexity. Please add a few sentences about the limitations of a scoping review. Page 7, Line 52 PsycInfo spelled incorrectly.
---

VERSION 1 – AUTHOR RESPONSE

We have reviewed all of the comments and have made revisions as described below. Changes to the manuscript are in blue with additional comments below.

1. Please include a PRISMA-P checklist, and fill it out with page numbers to indicate where relevant items have been included.

RESPONSE: This has now been uploaded as an additional document. SB has now been specified as the guarantor of the review (page 10, paragraph 5), and a plan for documenting important protocol amendments has now been included in the manuscript (page 10, paragraph 2).

2. The nature of the interventions to be scoped could be made more explicit. The purpose as stated in the abstract is to improve parent-child attachment, parental health, wellbeing, functioning or skills. The abstract could be clearer about whom the interventions might be targeted at (parents or child, whole families, or health or other service providers supporting parents) and when and where the intervention might take place. On page 5 it becomes clearer that interventions for staff or services that might impact on parent or family health are excluded – the intervention needs to involve the parents.

RESPONSE: The wording has now been clarified in the abstract and the main body of the manuscript (Page 5, Paragraph 1, highlighted) to make it clear that this will be a scoping review of parent and family based interventions to improve parental health, wellbeing, functioning or skills, that have the potential to be delivered within a hospital setting. The rationale for searching beyond hospital based interventions, in order to capture applicable and transferrable findings from other settings, has also now been made clearer (last sentence of the Background section, paragraph 1 page 5).

We have removed parent-child attachment from the abstract because, whilst this is an important area, we had excluded it from this review at an earlier date to provide a manageable scope and this was included in the abstract in error.

3. There is also some contradiction between the introduction and the section on the research question, where the focus seems to be on interventions that are healthcare based. The question (3) states routine health care contact. However, the inclusion criteria on page 6 still leaves it unclear as to whether non-healthcare settings (eg schools or home) will be included. For example, there is a large literature on self-help groups, which use volunteers (as in the inclusion criteria) but take place outside the health service. Will these interventions be included?

RESPONSE: Consistent wording has now been used throughout the document to refer to parent and family based interventions (in blue throughout) that have the potential to be delivered in the context of hospital care. As described in the response to Q2, interventions which have been evaluated in other settings which are potentially applicable and transferable to this population within a hospital setting will be included. Self-help group interventions (for example) which meet the inclusion criteria would be included, as would other forms of volunteer and peer-support, in addition to interventions delivered by healthcare professionals. Reference to volunteers has now been removed from the inclusion criteria to avoid confusion, as all interventions will be included unless they meet the exclusion criteria.

4. The scoping review excludes studies where the parent has the long-term condition. In practice, a disproportionate number of parents of CSHCN themselves have disability or chronic conditions. Does this mean that interventions that improve parental functioning through addressing the parent's health needs (eg mental health care for parents) would be excluded? Is the scoping review focused on parental management of the child, rather than on parental needs or family functioning (as mentioned in the last paragraph on page 3)?

RESPONSE: This exclusion criterion was added after initial informal searches yielded a significant

number of studies where there parent's illness was the major focus of the study. We do not intend to exclude studies where a parent has a long-term condition alongside their child. We have removed this from the exclusion criteria because it is clear from the inclusion criteria that parents must have responsibility for a child with a special health care need. Interventions which address health (including mental health) issues in a parent of a child with a special health care need will be included as long as they meet the other inclusion criteria.

5. Search terms: One of the questions stated on page 5 is "do the interventions work". To determine effectiveness, it seems important to identify studies that involve randomised comparisons. I would expect to see a more complex search strategy to achieve this.

RESPONSE: We have followed the Joanna Briggs methodology and have described a two stage process to develop the final search strategy, informed by a systematic review specialist who is a co-author (DB) and will be fully involved in during search strategy development. We have indicated in Table 3 the initial search terms that we intend to use, including study designs associated with randomised comparisons. Our initial phrasing was not clear because we stated that we had developed a search strategy which we have not yet done, we have now made it clear that we will develop a search strategy initially based on the terms in Table 3. The final search strategy will be published in the scoping review report.

6. The section on data analysis does not include any analysis of the strength of the evidence to answer a specific question. This could be a lengthy process rather like a systematic review. However, a summary that at least captures study design is needed to indicate areas of strengths and gaps in evidence.

RESPONSE: We have clarified within the research question section that we will look for evidence of efficacy or comparative effectiveness (page 5, paragraph 2). We have added study design to the list of key information to be charted, and stated that the risk of bias in controlled trials which contain comparative information on effectiveness will be appraised using conventional systematic review methods. (page 9, paragraph 2).

7. Abstract para 1 last line be clearer about what is meant by "this context"

RESPONSE: We have now specified that the context is a medically complex child's hospital admission and hospital care.

8. Methods/design: I think that it would be helpful to include a brief description of what formal scoping review methodology entails.

RESPONSE: A brief description of our intended methodology is now included (page 5, paragraph 3).

9. Search strategy: Please explain/justify why you will include grey literature in your search.

RESPONSE: We have now explained that we will include grey literature in the search in order to capture descriptions of relevant interventions which may not have been published in peer-review journals (page 8, paragraph 1).

10. A few suggestions for additional search terms

Parents: Include "guardian"

Child: Include “adolescent”

Interventions: Include “meta-analysis”

RESPONSE: Thank you, we have now included these terms (page 8, table 3).

11. Data analysis and synthesis; page 9; line 23: “Data will be coded according to predetermined codes derived from the research questions”. Please provide a brief example of this process.

RESPONSE: We have provided a brief description with examples of the process we intend to use (page 9, paragraph 3).

12. Please add a few sentences about the limitations of a scoping review.

RESPONSE: We have now included a paragraph on limitations (page 5 paragraph 4).

13. Page 7, Line 52 PsycInfo spelled incorrectly.

RESPONSE: Thank you, this is now corrected.

VERSION 2 – REVIEW

REVIEWER	Michael Takagi Murdoch Children's Research Institute Australia
REVIEW RETURNED	26-Jun-2017

GENERAL COMMENTS	The authors have addressed my comments well. I wish them the best of luck in performing this important review.
--